# Transport Characteristics of CJMAED™ Homogeneous Anion Exchange Membranes in Sodium Chloride and Sodium Sulfate Solutions

**DOI:** 10.3390/ijms22031415

**Published:** 2021-01-31

**Authors:** Veronika Sarapulova, Natalia Pismenskaya, Valentina Titorova, Mikhail Sharafan, Yaoming Wang, Tongwen Xu, Yang Zhang, Victor Nikonenko

**Affiliations:** 1Membrane Institute, Kuban State University, 149 Stavropolskaya St., 350040 Krasnodar, Russia; vsarapulova@gmail.com (V.S.); n_pismen@mail.ru (N.P.); valentina.titorova@mail.ru (V.T.); shafron80@mail.ru (M.S.); 2CAS Key Laboratory of Soft Matter Chemistry, Collaborative Innovation Center of Chemistry for Energy Materials, School of Chemistry and Material Science, University of Science and Technology of China, Hefei 230026, China; ymwong@ustc.edu.cn (Y.W.); twxu@ustc.edu.cn (T.X.); 3School of Environmental and Safety Engineering, Qingdao University of Science and Technology, 53 Zhenzhou Road, Qingdao 266042, China; zhangyang@qust.edu.cn

**Keywords:** anion exchange membrane, electric conductivity, diffusion permeability, permselectivity, structure–properties relationship, modification

## Abstract

The interplay between the ion exchange capacity, water content and concentration dependences of conductivity, diffusion permeability, and counterion transport numbers (counterion permselectivity) of CJMA-3, CJMA-6 and CJMA-7 (Hefei Chemjoy Polymer Materials Co. Ltd., China) anion-exchange membranes (AEMs) is analyzed using the application of the microheterogeneous model to experimental data. The structure–properties relationship for these membranes is examined when they are bathed by NaCl and Na_2_SO_4_ solutions. These results are compared with the characteristics of the well-studied homogenous Neosepta AMX (ASTOM Corporation, Japan) and heterogeneous AMH-PES (Mega a.s., Czech Republic) anion-exchange membranes. It is found that the CJMA-6 membrane has the highest counterion permselectivity (chlorides, sulfates) among the CJMAED series membranes, very close to that of the AMX membrane. The CJMA-3 membrane has the transport characteristics close to the AMH-PES membrane. The CJMA-7 membrane has the lowest exchange capacity and the highest volume fraction of the intergel spaces filled with an equilibrium electroneutral solution. These properties predetermine the lowest counterion transport number in CJMA-7 among other investigated AEMs, which nevertheless does not fall below 0.87 even in 1.0 eq L^−1^ solutions of NaCl or Na_2_SO_4_. One of the reasons for the decrease in the permselectivity of CJMAED membranes is the extended macropores, which are localized at the ion-exchange material/reinforcing cloth boundaries. In relatively concentrated solutions, the electric current prefers to pass through these well-conductive but nonselective macropores rather than the highly selective but low-conductive elements of the gel phase. It is shown that the counterion permselectivity of the CJMA-7 membrane can be significantly improved by coating its surface with a dense homogeneous ion-exchange film.

## 1. Introduction

Over the last 10 years, the number of publications aimed at developing new and improving existing anion exchange membranes (AEMs) has doubled and reached more than 650 articles per year (Scopus). There are several reasons for this interest in AEMs.

The first is the active development of methanol fuel cells [1,2] and redox flow batteries, in particular vanadium flow batteries using AEMs [3]. To ensure high performance of these new devices, AEMs are required that have high hydroxyl ion conductivity, mechanical robustness and high resistance to aggressive media, in particular, the membranes must be stable in a strongly alkaline environment [4,5].

The second important area is the production of electricity from renewable sources using reverse electrodialysis [6]. Thin, mechanically resistant AEMs with very low electrical resistance are being developed [7] to make this method not only environmentally sound but also economically viable.

The third direction is the use of AEMs in the food and pharmaceutical industries for the demineralization of proteins [8], tartrate stabilization of wine [9], separation of valuable anions of citric, malic, lactic acids and other organic nutrients [10,11]. In this case, the anion-exchange membranes should have sufficiently large pores that do not cause steric hindrances for the transport of large, highly hydrated anions.

The impetus for the activation of the fourth direction was the development of selectrodialysis and metathesis electrodialysis [12,13]. They allow the separation of streams of the ions, the joint presence of which in the concentrate can cause precipitation. Solving this problem revived interest in electrodialysis as an effective method for concentrating reverse osmosis retentates [14], deactivating mine and waste water [11,15], producing high-quality drinking water, water for irrigated agriculture [16], recovery of phosphates from municipal wastewater, livestock liquid effluent and landfills for the production of cheap fertilizers [17] as part of combined baro-electromembrane technologies. AEMs, which are selective to singly charged anions (systems Cl^-^/SO_4_^2−^, for example) [18,19] and are capable of resisting fouling, are essential for this purpose [20].

At the same time, AEMs that were developed and actively used in the 20th century often do not meet these requirements. Their operation in intense current modes, alkaline solutions or solutions of highly hydrated electrolytes leads to the destruction of the ion-exchange matrix [21], inert filler [22] and/or the transformation of strongly basic fixed groups (quaternary amines) into weakly basic tertiary and secondary amines [23]. In addition, the widespread introduction of electrodialysis, dialysis, membrane bioreactors, fuel cells, and other methods has until recently been constrained by the relatively high cost of AEMs.

In order for the new generation of AEMs to meet the above requirements and not have drawbacks, a number of innovations are under development. AEMs with new fixed groups [24], ion-exchange matrices [25], cross-linking agents [26], inert fillers and reinforcing materials [27], as well as methods of membrane manufacturing [28,29] and modifications of their surfaces [6,18,19] are actively developing. These new approaches are summarized in reviews [6,30]. New AEMs of the CJMAED series developed by Hefei Chemjoy Polymer Materials Co. Ltd. (Hefei, China), have a fundamentally different ion-exchange matrix compared to conventional commercial membranes, such as Neosepta AMX (ASTOM Corporation, Japan) and AMH-PES (Mega a.s., Praha, Czech Republic). First of all, they are designed for transport of large, highly hydrated ions and have already shown their effectiveness in extraction of 5’-ribonucleotides from hydrolysate [23], in recovery of gamma-aminobutyric acid (GABA) from reaction mixtures containing salt [31], in purification of methylsulfonylmethane from mixtures containing salt [32], in defluoridation of tea infusion [33], in separation of soluble saccharides from the aqueous solution containing ionic liquids [34] in salt valorization process from high salinity textile wastewater [35], in concentration of the high-salinity solutions prior to being treated by an evaporative crystallizer [36]. At the same time, in some cases, for example, when ED concentrating lithium sulphate from primary resources, new CJMAED membranes demonstrate lower efficiency compared to the traditional Neosepta AMX membrane [37]. Although CJMAED membranes have already found numerous applications, knowledge of their properties is fragmentary. In order to more confidently predict their behavior in new applications, as well as to obtain a fundamental basis for further progress in the synthesis of new membranes, more detailed studies of the structure–property relationship for CJMAED membranes are required.

This work presents the results of a comprehensive study of transport characteristics (electric conductivity, diffusion permeability, transport numbers) of CJMA-3, CJMA-6 and CJMA-7 anion exchange membranes manufactured by the Hefei Chemjoy Polymer Material Co. Ltd. in sodium chloride and sodium sulphate solutions. An analysis of their behavior is carried out in comparison with the widely used Neosepta AMX and AMH-PES membranes. Differences in the structure of these membranes and the impact of these differences on their characteristics are examined. The interplay between the ion exchange capacity, water content, conductivity, diffusion permeability, and counterion transport numbers (counterion permselectivity) of the membranes is studied on the bases of the microheterogeneous model [38]. In addition, we report the results of a surface modification of the CJMA-7 membrane, which significantly improved its counterion permselectivity, that is, decreased undesirable coion transfer. The latter reduces current efficiency of electrodialysis and does not allow obtaining brines of high concentration reducing current efficiency of electrodialysis.

## 2. Theoretical Background

The microheterogeneous model [38] considers an ion-exchange membrane (IEM) as a two-phase medium (Appendix A). One of the phases is the gel phase, which comprises microporous regions consisting of a polymer matrix with fixed groups. The charge of these groups is counterbalanced by a charged solution containing mobile counterions and, to a lesser extent, coions. The filaments of the reinforcing cloth and the inert filler (if any) are also included in the gel phase. The elements of the gel phase are separated by intergel spaces filled with an electrically neutral solution; these are the central parts of the meso- and macropores, together with structural defects of the membrane. It is assumed that there is local equilibrium between the elements of the gel phase and electroneutral solution filling the intergel spaces. When the membrane is in equilibrium with an external solution, the internal electroneutral solution is assumed identical to the external one. This model is used by many researchers to determine the structure–transport properties relationship for IEMs (see, for example [39,40]). The microheterogeneous model gives a simple description of membrane transport characteristics as functions of a single set of parameters, which are the ionic diffusion coefficients ($\bar{D_{i}}$) in the gel and solution ($D_{i}$) phases, the (dimensionless) volume fractions of the gel phase ($f_{1}$) and intergel spaces ($f_{2}$, $f_{1}+f_{2}=1$), the Donnan equilibrium constant ($K_{D}$), membrane ion-exchange capacity (Q) and a parameter (α), which reflects the relative disposition of the gel and solution phases [38]: α = −1 refers to the case where the elements of the gel and solution phases are connected in series; α = 1 refers to the case where these elements are connected in parallel. The main equations of the microheterogeneous model are in the Appendix A (SM). The exchange capacities of the membrane and the gel phase ($\bar{Q}$) are connected by the relation $Q=f_{1}\bar{Q}$.

According to the microheterogeneous model [38], within the concentration range 0.1 $C_{iso}<C<10C_{iso}$ and under condition that parameter α is not too great (|α|≤0.2), the electrical conductivity of an IEM, $\kappa^{*}$, can be expressed as:(1)$$\kappa^{*}={\bar{\kappa }}^{f_{1}}\kappa^{f_{2}}$$

Here $C_{iso}$ is the isoconductance concentration (the concentration at which the conductivities of the membrane, $\kappa^{*}$, gel phase, $\bar{\kappa }$, and bathing solution, κ, are the same); $C_{iso}$ is close to 0.05 eq L^−1^ for conventional IEMs. Here C (eq L^−1^) is the equivalent electrolyte concentration ($C=\left|z_{1}\right|c_{1}=\left|z_{A}\right|c_{A}$, where subscripts *1* and *A* refer to the counterion and coion; *c_i_* (mol L^−1^) is the molar ion concentration).

In the range of relatively dilute solutions mentioned above, the coion concentration in the gel phase is quite low due to the Donnan exclusion effect. Therefore, the main contribution to the gel phase conductivity is made by the counterions, whose concentration is very close to the concentration of the fixed ions. In these conditions, the value of $\bar{\kappa }$ is nearly independent of the external concentration, C. Assuming that $\bar{\kappa }=\mathrm{const}$ and κ is proportional to C, we arrive at the following equation connecting $\kappa^{*}$ with C:(2)$$\log\kappa^{*}\approx f_{2}\log C+const$$

The linearity of the $\log\left(\kappa^{*}\right)$—log(C) dependence for conventional IEMs is confirmed in a number of publications [38,39,40]; the $f_{2}$ parameter can be found as the slope of the above dependence. However, this dependence is often determined in a range outside of that close to $C_{iso}$ where Equation (1) holds; parameter α can be rather high (>0.2). In these conditions, the $\log\left(\kappa^{*}\right)$—log(C) slope can differ from the true volume fraction of the intergel spaces. Namely, in the range of relatively elevated concentrations, C > 0.5 eq L^−1^, the membrane conductivity depends on parameter α. At elevated concentrations, the gel conductivity is much less than the solution conductivity: $\bar{\kappa }\ll \kappa$. When α is small and the series connection of the phases is dominant, the membrane conductivity is controlled by the low-conductive gel regions, and it is relatively small. When α is great and the parallel connection is dominant, the membrane conductivity is high, since it is controlled by the well-conductive intergel solution regions (Figure 1). Therefore, the higher α, the higher the $\log\left(\kappa^{*}\right)$—log(C) slope. In any case, the $\log\left(\kappa^{*}\right)$—log(C) slope characterizes the contribution of the electroneutral solution filling the intergel spaces into the membrane conductivity. If $C>C_{iso}$, the higher the slope, the higher this contribution. When there is no electroneutral solution in the membrane, the $\log\left(\kappa^{*}\right)$—log(C) slope will be minimal, the increase in $\kappa^{*}$ is caused only by the coions in the gel phase, where their concentration is very low due to the Donnan exclusion. Maximum slope corresponds to the free solution. Since in the case of near-isoconductance concentration, this slope is equal to $f_{2}$, we call the $\log\left(\kappa^{*}\right)$—log(C) slope the apparent fraction of the electroneutral solution in the membrane, $f_{2app}=d\log\kappa^{*}/d\log C$.

At C =$\;C_{iso}$, the conductivity of the gel phase, $\bar{\kappa }$, can be approximately (when neglecting the contribution of the coions, which is generally insignificant) presented as follows:(3)$$\bar{\kappa }=\frac{z_{1}{\bar{D}}_{1}\bar{Q}F^{2}}{RT},\;\;$$
(4)$$\bar{Q}=\frac{Q}{f_{1}},$$
where $\bar{Q}$ is the concentration of the fixed ions in the gel phase; $z_{1}$ is the counterion charge number, subscript 1 refers to the counterion; *F*, *R,* and *T* are the Faraday constant, absolute temperature, and gas constant, respectively.

The differential (or local) diffusion permeability coefficient, $P^{*}$, is defined from the transport equation deduced within the irreversible thermodynamics [38,41]:(5)$$j_{i}=-P^{*}\frac{dc_{i}}{dx}+\frac{it_{i}^{*}}{z_{i}F},$$
where *i* is the current density, $j_{i}$ and $t_{i}^{*}$ are the flux density and the transport number of ion *i* in the membrane, $c_{i}$ is the molar concentration of this ion in the “virtual” (or “corresponding” [41]) solution, which is an electrically neutral solution being in equilibrium with a thin membrane layer with coordinate *x*. The asterisk means that the quantity refers to a small membrane volume with coordinate *x*.

The differential and integral permeability coefficients are linked as follows [38]:(6)$$P^{*}=P+C\frac{dP}{dC}$$

Equation (6) can be presented in the form [42]:(7)$$P^{*}=P\left(1+\beta \right),$$
where β=dlogP/dlogC. Equation (7) is obtained from Equation (6) as follows: $P+C\frac{dP}{dC}==P\left(1+\frac{C}{dC}\frac{dP}{P}\right)=P\left(1+\frac{dlogP}{dlogC}\right)=P\left(1+\beta \right)$.

The value of $P^{*}$ can be found from the concentration dependence of *P* presented in bilogarithmic plot (logP vs. logC). The slope of this dependence gives β.

If the external solution concentration is not too high, an approximate form of the Donnan relation can be used to calculate the coion concentration in the gel phase: $\bar{c_{A}}=K_{D}c_{A}^{2}/\bar{Q}$ (written for a 1:1 electrolyte [38]). Using the last relation, an approximate formula to calculate $P^{*}$ in nonconcentrated solutions (up to 1 mol L^−1^) can be obtained:(8)$$P^{*}=2D_{A}t_{1}^{*}{\left[f_{1}{\left(K_{D}\frac{{\bar{D}}_{A}c_{A}}{D_{A}\bar{Q}}\right)}^{\alpha }+f_{2}\right]}^{1/\alpha }$$

Thus, the membrane diffusion permeability is controlled by the coion diffusion in the gel phase and in the intergel spaces. The former is determined by the parameter $G=K_{D}{\bar{D}}_{A}/\bar{Q}D_{A}$, first proposed by Gnusin [43]. This parameter reflects both the ability of the gel phase to coion sorption (the $K_{D}/\bar{Q}$ ratio) and the coion mobility in this phase (the ${\bar{D}}_{A}/D_{A}$ ratio). The electrolyte diffusion through the intergel solution is controlled by the $f_{1}$ ($f_{2}$) and α parameters, as well as by $D_{A}$. When applying parameter G, only four parameters are needed to describe the transport properties of IEMs: $\bar{Q}$, G, $f_{1}$ ($f_{2}$), and α. To find them by fitting the experimental concentration dependencies of $\kappa^{*}$ and $P^{*}$, we use the feature that $f_{2}$ mainly determines the slope of the $\log\left(\kappa^{*}\right)$—log(C) curve, while $\bar{Q}$ shifts the curve up or down; a similar role is played by α and G when treating $\log\left(P^{*}\right)$—log(C) curve.

The values of $\kappa^{*}$ and $P^{*}$ allow determination of the transport numbers of counterions ($t_{1}^{*}$) and coions ($t_{A}^{*}$) in an IEM according to an approximate relation [44]:(9)$$t_{1}^{*}=\frac{1}{2}+\sqrt{\frac{1}{4}-\frac{\left(z_{1}\left|z_{A}\right|\right)P^{*}F^{2}C}{\left(z_{1}+\left|z_{A}\right|\right)RT\kappa^{*}}},\;t_{A}^{*}=1-t_{1}^{*}$$

## 3. Membranes

### 3.1. Manufacturing

Conventionally, ion-exchange membranes are classified as homogeneous and heterogeneous. Heterogeneous membranes (which historically were produced earlier) are prepared by using powdered ion-exchange resin, which is mixed with a powder of a filler, such as polyethylene. The ion-exchange material (the particles of ion-exchange resin) is unevenly and sometimes discontinuously distributed in the membrane. Homogenous membranes are prepared by introducing an ion-exchange moiety directly into the structure of the constitutive polymer [45]. In this case, the ion-exchange material forms a continuous phase, which is relatively evenly distributed in the membrane. The reinforcing cloth (tissue, fabric) is added to both homogeneous and heterogeneous membranes. There are a few types of membranes (such as some marques of Nafion), which do not contain reinforcing cloth.

The commercial CJMA-3, CJMA-7 [46] and experimental CJMA-6 homogeneous AEMs are manufactured by the Hefei Chemjoy Polymer Materials Co. Ltd. (Hefei, China). The generic name of this type of membrane CJMAED on the manufacturer’s website emphasizes their possible use in electrodialysis. Their ion-exchange matrix contains polyvinylidene fluoride, PVDF (CJMA-3), or polyolefin (CJMA-6, CJMA-7) functionalized with quaternary ammonium groups [31]. The matrix is crosslinked through the side chains [47]. These membranes are produced by the casting method and are reinforced with polyethylene terephthalate, PET (trademarks: Terylene, or Lavsan, or Dacron) cloth by hot rolling. The CJMAED membranes differ from each other in the type of polymer matrix, in the degree of crosslinking and in the concentration of fixed groups. These membranes are less costly compared with other anion-exchange membranes on the world market.

The Neosepta AMX (ASTOM Corporation, Tokyo, Japan) and Ralex AMH-PES (Mega a.s., Czech Republic) anion exchange membranes have a highly crosslinked aromatic ion exchange matrix based on copolymer of polystyrene (PS) and divinylbenzene (DVB). These membranes also mainly contain quaternary ammonium groups. The pseudohomogeneous Neosepta AMX membrane is produced using the “paste method” [48]. The reinforcing PVC cloth as well as PVC particles with diameter less than 60 nm (inert filler) are introduced into the membrane at the stage of the ion-exchange matrix polymerization [49]. The AMH–PES heterogeneous membrane is produced by hot rolling of the mixture of a fine powder of the ion-exchange resin Lewatit M500 with an average diameter of about 5 microns and low density polyethylene (PE), at an approximate ratio of 2:1 [50]. A reinforcing Ulester 32S net is rolled into both sides of the AHM-PES membrane while the membrane is still hot [51,52]. The data on the membranes under study are summarized in Table 1.

### 3.2. Surface and Cross-Section Visualization

Figure 2 and Figure 3 demonstrate optical images of the surfaces and cross-sections of swollen AEMs. The obtained AMX and AMH-PES images are in good agreement with the known results of numerous studies of their structure, presented, for example, in [52,53,54]. Reinforcement cloth is evenly distributed over the cross-section of the AMX membrane (Figure 2a). Both surfaces of this membrane are wavy. The characteristic distance between the “peaks” and “valleys” of the AMX surface relief is determined by the periods of undulation of the filaments of reinforcing cloth and is 250 μm. The difference between the highest and lowest points of the waves is 15 ± 5 µm. The AMX membrane has practically no macropores [55] due to the good adhesion between the PVC reinforcing cloth and the ion exchange composite, which contains PVC as an inert filler.

Reinforcing cloth is located near both surfaces of the heterogeneous AMH-PES membrane (Figure 2b) and the threads sometimes jutted from the ion-exchange material [51]. These protrusions, as well as those of the ion exchange resin particles, are responsible for the surface roughness of the AMH-PES. The difference between the highest and lowest points of the AMH-PES surface is 15 ± 2 µm. Macropores are mainly localized at the ‘ion exchange resin/PE’ and ‘reinforcing cloth/composite ion exchange material’ boundaries [56]. The reinforcing cloth of all CJMAED membranes is localized closer to one of the surfaces (surface I). As a result, surface II is smoother and surface I is wavier (Figure 3). Moreover, the bulges are localized over the intersections of the threads of the reinforcing cloth. In these places the filaments sometimes protrude above the ion-exchange material. The distance between two neighboring highest points of the surface is determined by the cell step of the reinforcing cloth, equal to 250 μm. Spread of heights of the surface I can reach 80 μm that significantly exceeds the undulation of the AMX surface. The difference in the geometric characteristics of surfaces I and II for the studied CJMAED membranes increases with increasing membrane thickness in the sequence: CJMA-6 < CJMA-3 < CJMA-7. Note that the optical images of the swollen CJMA-3 membrane surfaces are in good agreement with its SEM images obtained for dry samples [56].

Extended macropores are localized at the boundaries between the filaments of reinforcing cloth and the ion-exchange material of CJMAED membranes. These macropores are visualized as light stripes in images of the swollen sample exposed to air with surface I (Figure 4). They appear at the crosshairs of the reinforcing filaments that are closest to the surface during the first few seconds of drying of the test sample. As it dries, these light stripes spread along the entire length of the reinforcing filaments. As will be shown in Section 5, the presence of such macropores, that are in contact with the external solution due to the protrusions of the reinforcing cloth filaments to the surface, can strongly affect the transport characteristics of CJMAED membranes.

## 4. Membranes Characterization

### 4.1. Basic Characteristics

Table 2 summarizes some of the characteristics of the studied AEMs that we obtained or found in the literature.

The thickness of the swollen heterogeneous AMH-PES membrane is several times greater than that of the swollen homogeneous membranes, which increase in the following order: CJMA-6 < AMX < CJMA-3 < CJMA-7. The exchange capacities of swollen membranes (*Q*) having an aromatic matrix (AMX, AMH-PES) significantly exceed the values found for CJMAED membranes, for which *Q* increases in the series: CJMA-3 < CJMA-7 < CJMA-6. Concentration of fixed groups (the number of millimoles of fixed groups per 1 cm^3^ of water in the membrane), *C_X_*, grows in the following order: CJMA-7 <<CJMA-3 ≈ AMH-PES < CJMA-6 < AMX. The difference in the sequences found for *C_X_* and *Q* is caused by a twofold increase in the water content in the AMH-PES and CJMA-7 membranes compared to the other membranes under study. As expected, the density of the highly hydrated AMH-PES and CJMA-7 membranes approaches that of water. The density of the less hydrated AMX, CJMA-3 and CJMA-6 membranes is more dependent on the density of the polymers [57] from which they are made. The thickest AMH-PES membrane has the highest electrical resistance in a moderately concentrated (0.5 M) NaCl solution. In the next section, we will try to explain the order in which the electrical resistance of homogeneous membranes increases: CJMA-7 < AMX ≈ CJMA-3 < CJMA-6.

### 4.2. Transport Properties

#### 4.2.1. Electrical Conductivity

As already mentioned in Section 2, Equation (2), which was obtained in the framework of the microheterogeneous model, is fulfilled in the concentration range 0.1$C_{iso}<C<10C_{iso}$, where $C_{iso}$ is the concentration isoelectric conductivity. From Equations (2) and (3) it follows: near $C_{iso}$, the electrical conductivity of the membrane in the first approximation is determined by the exchange capacity of the gel phase (the concentration of fixed groups $\bar{Q}$) and the mobility of counterions in this phase, equal to $z_{1}{\bar{D}}_{1}F/RT$. Note that the higher the water content of the membrane per fixed group, the higher the mobility. The higher $\bar{Q}$, the higher the electrical conductivity of the gel phase, $\bar{\kappa }$, and, accordingly, that of the membrane, $\kappa^{*}$. With an increase in C, the value of $\bar{\kappa }$ changes insignificantly, mainly due to the sorption of coions. At the same time, the conductivity κ of the intergel space (the central part of the pores) filled with a solution identical to the external one increases nearly proportional to C. With increasing C, the effect of the values of κ, $f_{2app}$, and α on the membrane electrical conductivity rises. As mentioned above, the value of $f_{2app}$ increases with increasing concentration. It is due to increasing contribution of the intergel solution to the overall membrane conductivity; in addition, in the range of elevated concentrations, coions are more and more involved in the conductivity of the gel phase because of their increasing sorption by the gel. The properties of membranes differ most of all in the region of high concentrations. For these reasons, the $f_{2app}$ values were determined in the 0.4–1.0 eq L^−1^ concentration range, where 1.0 eq L^−1^ is the maximum concentration used in the study.

Figure 5 shows concentration dependencies of the conductivity of AEMs in NaCl and Na_2_SO_4_ solutions. Table 3 summarizes the values of the membrane parameters found from these dependencies along with the data on the membrane diffusion permeability measured as the functions of concentration (Section 4.2.2). These parameters are the input parameters for the microheterogeneous model [38]. The values of these parameters were found by fitting the results of modelling to the experimental data. We have used the following features of the model behavior: when fitting the *κ** vs. C dependence, the *f*_2_ value controls the slope of the log*κ** vs. logC curve, while ${\bar{D}}_{Cl^{-}}$ shifts the curve up (when ${\bar{D}}_{Cl^{-}}$ increases) or down (when it decreases); when fitting the *P** vs. *C* dependence, similarly, the slope and position of the log*P** vs. logC may be changed by varying the values of parameters α and $G=K_{D}{\bar{D}}_{Na^{+}}/(\bar{Q\;}D_{Na^{+}})$. Generally, there is a good agreement with the data known in the literature. As Figure 5 shows, the difference between the calculated and experimental results does not exceed the experimental error.

The AMH-PES membrane is a heterogeneous one, hence it contains macropores. In addition, this membrane is relatively highly hydrated (Table 2), which leads to a higher ion mobility in the gel phase and a lower effect of Donnan coion exclusion (Table 3). As a result, the values of ${\bar{D}}_{Cl^{-}}$ and G for this membrane are elevated, and the $f_{2app}$ value is about 1.5 times greater than that for the homogeneous AMX membrane. In the studied concentration range ($C>C_{iso}$) of NaCl solutions, the conductivity of AMH-PES is significantly higher than the AMX conductivity (Figure 5a). Moreover, in accordance with the microheterogeneous model, this difference increases with increasing concentration of the external solution.

The CJMAED membranes have elevated values of $f_{2app}$. At concentrations of the external solution close to $C_{iso}$, the conductivity, according to Equation (3), is determined by the product of $\bar{Q}$ and ${\bar{D}}_{Cl^{-}}$. The latter mainly depends on the water uptake by the gel phase, the maximum is for the CJMA-7 membrane, accordingly, the counterion mobility in the gel phase of this membrane is the greatest one among the studied membranes. The ${\bar{D}}_{Cl^{-}}$ values are not very different for the CJMA-3 and CJMA-6 membrane, but $\bar{Q}$ is higher for the latter, which determines its higher conductivity compared to the CJMA-3 membrane. With an increase in the concentration of the external solution, the conductivity of CJMA-6 grows faster than the conductivity of CJMA-3, but the difference between them remains small. At the same time, the difference between the conductivities of CJMA-6 and AMH-PES membranes decreases with increasing concentration due to the fact that the $f_{2app}$ value for CJMA-6 is about two times higher than that for AMH-PES. The highest values of conductivity (and the lowest values of electrical resistance, Table 2) in the studied range of NaCl concentrations are demonstrated by the CJMA-7 membrane, for which the $f_{2app}$ parameter is about 0.35. Note that $f_{2app}$ for this membrane is significantly higher than that for the heterogeneous MK-40 and MA-41 membranes ($f_{2app}$ = 0.21 ± 0.03) [71]. This experimental fact allows us to conclude that the CJMA-7 membrane contains an ion-exchange material, which has more large pores in comparison with other membranes under study. The dimensions of the pores in the ion-exchange materials of the CJMA-3 and CJMA-6 membranes are apparently significantly smaller.

Replacing the singly charged Cl^-^ counterion with the doubly charged SO_4_^2−^ counterion results in a decrease in the conductivity of all studied AEMs (Figure 5b). Estimates made using the microheterogeneous model [38] allow us to conclude that in the gel phase of AMX, AMH-PES and CJMA-3 membranes, the ratio of diffusion coefficients of doubly charged and singly charged counterions is 0.3–0.4 (Table 3), while in solution at infinite dilution (Appendix A, Appendix A) $D_{SO_{4}^{2-}}/D_{Cl^{-}}\;$= 0.5. In the literature, such a decrease is explained by lower mobility of doubly charged counterions as a result of ion–ion interactions simultaneously with two fixed groups [72,73] and/or steric hindrances caused by the large size of strongly hydrated ions [74,75], which are sulfate ions (Appendix A, Appendix A). Low values of the conductivity of the Na_2_SO_4_ solution in the intergel spaces also contributes to the lower values of the conductivity of the membranes in this solution.

In the case of CJMA-6 and CJMA-7 membranes, which are characterized by the highest $f_{2app}$ values, the ratio ${\bar{D}}_{SO_{4}^{2-}}/{\bar{D}}_{Cl^{-}}$ increases. This means that the structural features of the ion-exchange matrix of these membranes make it possible to at least partially remove the restrictions on the transport of sulfate ions in comparison with chloride ions. This property of the new CJMA-6 and CJMA-7 membranes is very attractive for their use in the electrodialysis processing of sulfate and other solutions containing large anions.

#### 4.2.2. Diffusion Permeability

The concentration dependences of the integral coefficient of diffusion permeability, *P* , for the studied AEMs, as well as the counterion transport numbers in these membranes are shown in Figure 6 and Figure 7, respectively.

In the case of the CJMA-7 membrane, the values of P in NaCl solution are several times higher than the corresponding value for the AMH-PES membrane and are more than one order of magnitude higher than the P for the AMX, CJMA-6 and CJMA-3 membranes. Apparently, the dominant role is played by the diffusion of electrolyte through the macropores of the CJMA-7 membrane. Three parameters contribute to the diffusion permeability according to the microheterogeneous model [38]: $\bar{Q}$, $f_{1}$ ($f_{2}$) and α.

The influence of these parameters on the differential diffusion coefficient of AEMs, $P^{*}$ (see the definition of $P^{*}$ in Section 2), can be expressed by the following equation [55]:(10)$$P^{*}={\left\{{\left[f_{1}{\left(\frac{\bar{P}}{\bar{t_{1}}}\right)}^{\alpha }+f_{2}{\left(\frac{D}{t_{1}}\right)}^{\alpha }\right]}^{-1/\alpha }+{\left[f_{1}{\left(\frac{\bar{P}}{\bar{t_{A}}}\right)}^{\alpha }+f_{2}{\left(\frac{D}{t_{A}}\right)}^{\alpha }\right]}^{-1/\alpha }\right\}}^{-1},$$
where ${\bar{t}}_{1}$ and $t_{1}$, as well as ${\bar{t}}_{A}$ and $t_{A}$ are the counterion and coion transport numbers in the gel phase and solution, respectively. The diffusion permeability of the gel phase for 1:1 electrolyte (NaCl) and 2:1 electrolyte (Na_2_SO_4_) can be approximated as [55]:
(11)$$\bar{P}=2\bar{t_{1}}{\bar{D}}_{A}K_{D}\left(\frac{C}{\bar{Q}}\right),\;\left|z_{1}\right|=\left|z_{A}\right|=1$$
(12)$$\bar{P}=\frac{3}{2}\bar{t_{1}}{\bar{D}}_{A}K_{D}{\left(\frac{C}{\bar{Q}}\right)}^{1/2},\;\left|z_{1}\right|=2,|z_{A}|=1$$
where C is the equivalent concentration of external solution ($C=z_{i}c_{i}$).

The AMX and AMH-PES membranes have close values of $\bar{Q}$ (Table 3) and similar ion exchange matrices based on DVB/PS (Table 1 and Table 3). The similarity of the matrices predetermines close values of α (Table 3). Therefore, as predicted by the model, the diffusion permeability of these membranes increases with growth of external solution NaCl concentration in order AMX < AMH-PES according to the increase in the volume fraction $f_{2}$ of the intergel solution. The CJMA-6 and CJMA-7 membranes have close values of $f_{2}$, but different values of $\bar{Q}$. According to Equations (11) and (12), $\bar{P}$ is inversely proportional to $\bar{Q}$: the higher the concentration of fixed groups, the stronger the Donnan exclusion of coions from the gel phase [72]. A very low permeability of the gel phase of the CJMA-6 membrane is characterized by an extremely small value of parameter G (Table 3).

The value of α plays also an important role, which influences the slope of $\log\left(P^{*}\right)$ - log(C) curve: the greater α, the greater the value of $P^{*}$ (but the lower the slope). The highest rate of diffusion can be obtained when α = 1 and the electrolyte freely passes across the through pores. With decreasing α, the electrolyte needs to overcome more and more of the gel phase elements, which are significant barriers to coions. Among the three CJMAED membranes, the CJMA-7 membrane has the highest values of α; as well, the values of $f_{2}$ and $f_{2app}$ are the highest for this membrane. In addition, the value of G, which characterizes the ability of coions to cross the gel phase, is especially great. As a result, the value of $P^{*}$ for this membrane is only 25 times lower than the NaCl diffusion coefficient in solution. For the CJMA-3 membrane this factor is about 250, and for the CJMA-6 and AMX membranes it is equal to 750 (all values are given for the case of 1 eq L^−1^ NaCl solution). The importance of parameters α and G also follows from the fact that the value of $P^{*}$ for the CJMA-6 membrane is significantly lower than for the AMX, despite the higher $f_{2}$ (CJMA-6) and the close $\bar{Q}$ (Table 3).

The replacement of a NaCl solution with a Na_2_SO_4_ solution with the same normality leads to a noticeable increase in the diffusion permeability of all studied membranes (Figure 3). A similar trend was found for a Russian heterogeneous anion-exchange MA-41 membrane earlier [76]. From Equations (11) and (12) it follows that ${\bar{P}}_{NaCl}<{\bar{P}}_{Na_{2}SO_{4}}$, if we take into account that the value $C/\bar{Q}\;<{\left(C/\bar{Q}\right)}^{1/2}$, when $C/\bar{Q}$ is significantly less than unity. The increase in the diffusion permeability of the gel phase is caused by an increase in the coion concentration in this phase due to increasing electrostatic interactions between counterions and coions. Another reason for the increase in the diffusion permeability of membranes in Na_2_SO_4_ solutions can be a slight increase in $f_{2}$ ($f_{2app}$) (Table 3). The latter is caused by the stretching of the ion-exchange matrix [72] when highly hydrated sulfate ions (Appendix A, Appendix A) are introduced into its pores. A similar effect, confirmed by the data of standard contact porosimetry, was observed in our study of the diffusion permeability of AEMs in phosphate-containing solutions [55].

It follows also from Equations (11) and (12) that the value of $\bar{P}$ should increase with increasing concentration of the external solution. Indeed, the diffusion permeability of AEMs increases with increasing NaCl solution concentration (Figure 6a,b). At the same time, a slight increase in the diffusion permeability of AEMs in dilute Na_2_SO_4_ solutions is replaced by a decrease when increasing the Na_2_SO_4_ concentration (Figure 6c). This, at first glance, an unexpected feature in the shape of the P vs. C dependence, apparently, is caused by a sharp drop in the diffusion coefficient of Na_2_SO_4_ (D) with an increase in the solution concentration (Appendix A, Appendix A).

The transport numbers of the Cl^-^ (Figure 7a) and SO_4_^2+^ counterions (Figure 7b), which characterize the selectivity of AEMs with respect to anion transport, decrease in a series that, according to Equation (9), corresponds to an increase in the $P^{*}/\kappa^{*}$ ratio. Higher diffusion permeability of membranes in Na_2_SO_4_ solutions and their lower conductivity in the SO_4_^2−^-form result in lower values of $t_{SO_{4}^{2-}}^{*}$ compared to $t_{Cl^{-}}^{*}$.

The AMX and CJMA-6 membranes demonstrate the highest and nearly the same counterion permselectivity in the investigated range of electrolyte concentrations. The selectivity of the AMH-PES and CJMA-3 membranes is noticeably lower, as mentioned above, due to either the higher values of $f_{2app}$ and α (AMH-PES), or the low exchange capacity (CJMA-3). The CJMA-7 membrane has the lowest selectivity, since it is characterized by the highest $f_{2app}$ and a sufficiently large α. Note that in moderately dilute (0.1 eq L^−1^) NaCl and Na_2_SO_4_ solutions, the $t_{1}^{*}$ value for all membranes, except for CJMA-7, is close to unity. Moreover, even in concentrated (1.0 eq L^−1^) NaCl and Na_2_SO_4_ solutions, the $t_{1}^{*}$ values for these membranes (except for CJMA-7) exceed 0.95. This means that these membranes can be used in electrodialysis for both demineralization and concentration of solutions.

Below we show that the selectivity of the CJMA-7 and similar membranes with a high $f_{2app}$ value can be improved by at least partially eliminating structural defects caused by the embedding of the reinforcing cloth (see Section 3.2).

## 5. Increasing CJMA-7 Permselectivity by Surface Modification

Optical images (Figure 3) allow us to see some filaments of CJMA-7 reinforcing cloth protruding to the surface. This causes formation of large macropores whose mouths open into the external solution. These large and long pores facilitate the access of the external solution into the membrane bulk. We assume that this defect is one of the main reasons for the low permselectivity of the CJMA-7 membrane. We try to reduce this defect by co-vering the membrane surface with a dense ion-exchange film. For this purpose, we used a sample designated as CJMA-7’, which is the CJMA-7 membrane that was stored in a refrigerator at a temperature of +8 °C in a 0.1 N NaCl solution for 10 months.

A 4 μm thick film of sulfonated fluoropolymer MF-4SK was then formed on the surface of the CJMA-7’ sample. The CJMA-7’M-25 and CJMA-7’M-50 samples were obtained according to the procedure described in Section 6.2. A sample of the CJMA-7’ membrane, which was used for comparison, was in the air for the same time (24 h) and at the same temperature (+25 °C) as the sample CJMA-7’M-25 during its preparation. Sample CJMA-7’M-50 was kept at 50 °C for the last 1 h of modification. Figure 8 shows the scheme of the treatment applied to the CJMA-7 membrane.

Figure 9 shows the concentration dependences of the conductivity of the pristine membrane and modified samples. Figure 10 shows the diffusion permeability and coion transport numbers ($t_{A}^{*}$) in the original membrane and modified samples. The data were obtained in NaCl solutions.

A prolonged exposure of the CJMA-7 membrane at a relatively low (8 °C) temperature for a long time resulted in a slight decrease in electrical conductivity (Figure 9b), as well as in an almost twofold decrease in the diffusion permeability (Figure 10a) and more than twofold decrease in the coion transport number (Figure 10b) compared to the pristine membrane. These changes are apparently caused by the contraction of the weakly cross-linked ion exchange matrix during storage of this membrane. It is known, for example, that in the case of Nafion materials, the relaxation of their aliphatic matrix after exposure to temperature takes 1000 or more hours [77]. From the data presented in Figure 9 and Figure 10 it can be seen that the application of a film on the CJMA-7’ membrane results in a more significant reduction (by about 60%) in the conductivity. At the same time, the diffusion permeability of CJMA-7’ decreased even more significantly, which led to a noticeable decrease in the coion transport numbers: by 10% in the case of drying at 25 °C (the CJMA-7’M-25 membrane) and by 40% in case of drying at 50 °C (the CJMA-7’M-50 membrane).

Thus, the transport number of Cl^−^ ions in the CJMA-7’M-50 membrane in 1 eq L^−1^ NaCl is approximately equal to 0.97, which places this membrane in the counterion permselectivity between AMH-PES and AMX (Figure 7). The modifying film clogs the outlets of extended macropores on the membrane surface. As a result, the $f_{2app}$ value found from the concentration dependences of the electrical conductivity decreases from 0.33 (CJMA-7’) to 0.28 (CJMA-7’M-25, CJMA-7’M-50). A decrease in the conductivity of the modified samples in the range of relatively high concentrations is explained by a decrease in the volume fraction of a highly conductive electrically neutral solution in the membrane due to the structural changed caused by the modification of the pristine membrane. Note that the conductivity of the modified CJMA-7’M samples in 0.1 eq L^−1^ NaCl remains very high, about 6 mS cm^−1^, which is approximately equal to the conductivity of AMH-PES (Figure 4) and is noticeably higher than the conductivity of AMX (4 mS cm^−1^). The value of diffusion permeability decreased to a much greater extent (about six times, when comparing the CJMA-7 and CJMA-7M-50 membranes). The main cause is the decrease in the volume fraction of the intergel electroneutral solution, which significantly reduces the possible routes of coion transport through the membrane. A strong decrease in the diffusion permeability leads to a three-fold decrease in the coion transport number (Figure 10b).

## 6. Materials and Methods

### 6.1. Solutions

Solid NaCl and Na_2_SO_4_ of the analytical grade (Vecton JSC, St. Petersburg, Russia) as well as distilled water with electrical conductivity of 1.1 ± 0.1 μS cm^−1^ and pH = 5.5 (25 °C) were used for preparation of solutions. These solutions have a pH 5.4 ± 0.3 (NaCl) and 5.6 ± 0.3 (Na_2_SO_4_). Some characteristics (at 25 °C) of the NaCl and Na_2_SO_4_ solutions as well as the ions they contain [78,79] are represented in Appendix A, Appendix A. Concentration dependences of NaCl [78] and Na_2_SO_4_ [80] diffusion coefficients (at 25 °C) normalized to the diffusion coefficient at infinite dilution solution (Appendix A) are shown in Appendix A (see Appendix A)

### 6.2. Membrane Modification

For a year, the CJMA-7 membrane was stored in a refrigerator at a temperature of +8 °C in a 0.1 eq L^−1^ NaCl solution (sample CJMA-7’). Then the CJMA-7’ membrane was divided into two samples. A 7% solution of MF-4SK (perfluorinated sulfonated polymer in isopropyl alcohol, manufactured by Plastpolymer, Russia) [81] was poured onto the surface of each sample and dried at room temperature for 24 h until the MF-4SK film was formed (CJMA-7’M). Then these samples were dried for an additional 1 h at temperatures of 25 (sample CJMA-7’M-25) and 50 °C (sample CJMA-7’M-50) in an oven (Binder ED Avantgarde.Line, BINDER GmbH, Germany). During all the time (25 h) the CJMA-7’ sample was in air at room temperature (25 °C) in order to exclude from consideration possible changes in the characteristics of CJMA-7’M-25 and CJMA-7’M-50 caused by the presence of membranes in the air.

### 6.3. Study of Membrane Characteristics

All studied AEMs underwent salt pretreatment before experiments [82]. The characteristics of these membranes, presented in Table 2, were found by standard methods [83], the description of which is in Appendix A.

Surface and cross-section visualization of swollen AEMs was carried out with the SOPTOP CX40M optical microscope (Yuyao, Zhejiang, China) with a digital eyepiece USB camera (5×, 10×, 20×, and 50× magnification).

The AEMs electrical conductivity ($\kappa^{*}$) was measured by the differential method using a clip cell [84] and an immittance meter AKIP 6104 (B + K Precision Taiwan, Inc., New Taipei City, Taiwan) at an alternating current frequency of 1 kHz.

The confidence interval of the determination of the $\kappa^{*}$ value is equal to 0.4 mS cm^−1^.

The integral diffusion permeability coefficient of AEMs (P) was obtained using a two-compartment flow cell [85]. Appendix A contain the scheme (Appendix A) of the cell, as well as the details of the measurements and data processing. The confidence interval of the determination of (P) is equal to 0.4 × 10^−8^ cm^2^ s^−1^.

## 7. Conclusions

In moderately dilute (0.1 eq L^−1^) NaCl and Na_2_SO_4_ solutions, the counterion permselectivity of AMX, AMH-PES and CJMA-6, CJMA-3 membranes (quantified by the values of the counterion transport numbers $t_{1}^{*}$) is close to unity. Moreover, even in concentrated (1.0 eq L^−1^) NaCl and Na_2_SO_4_ solutions, the $t_{1}^{*}$ values for these membranes do not fall below 0.95 ± 0.01. This means that the listed AEMs can be used both for electrodialysis desalination and concentration of electrolyte solutions.

Structural parameters provide the CJMA-6 and CJMA-3 membranes with a rather low diffusion permeability in dilute solutions, despite their relatively low exchange capacity. A feature of the CJMA-6 membrane is that it has a relatively high portion of series-connected elements of a highly selective gel phase and intergel spaces filled with an electrically neutral solution (assumed identical to the external solution). Another particularity of this membrane is its fairly dense gel phase, which is very low permeable to coions. This determines the low diffusion permeability and high counterion permselectivity of this membrane. For these characteristics, the CJMA-6 membrane is not inferior to the AMX membrane, which is considered as one of the best in the world market. In the range of relatively high concentrations (about 1 eq L^−1^), the conductivity of the CJMA-6 membrane is approximately equal to the conductivity of AMX membrane. At the same time, in the range of low concentrations $\kappa_{CJMA-6}^{*}<\kappa_{AMX}^{*}$, which is explained by the higher exchange capacity and lower volume fraction of electroneutral solution in the AMX membrane. As for the CJMA-3 and CJMA-7 membranes, a relatively high contribution of the parallel connection of elements of the gel phase and intergel solution facilitates the electrolyte transfer through these membranes, which significantly reduces their counterion permselectivity.

The influence of the high volume fraction of the intergel solution and the contribution of the parallel connection of the gel phase and intergel solution on the diffusion permeability of CJMAED membranes is most pronounced in Na_2_SO_4_ solutions. This is due to two phenomena: (1) an increase in the linear dimensions of the pores of weakly crosslinked polymers due to stretching of the ion-exchange matrix when highly hydrated sulfate ions are introduced into its pores, and (2) a sharp decrease in the mobility of sulfate ions in solution with an increase in its concentration.

The CJMA-7 membrane has the lowest exchange capacity and the highest volume fraction of intergel spaces filled with the external solution. The gel phase of this membrane appears to be relatively porous, which allows both voluminous counterions and coions to pass through it. These properties determine the lowest selectivity of CJMA-7 in comparison with other investigated AEMs, which nevertheless does not fall below 0.87 ± 0.01 even in 1.0 eq L^−1^ NaCl and Na_2_SO_4_ solutions. It is preferable to use this membrane in the electrodialysis processing of dilute solutions, as well as for processing solutions containing large ions, for example, in the food industry.

One of the main reasons for a high diffusion permeability and low permselectivity of CJMA-7 is the presence of extended macropores, which are formed at the boundaries of ion-exchange material and the reinforcing cloth filaments. This defect is amplified when the reinforcing filaments are protruded to the membrane surface. It is shown that the CJMA-7 membrane permselectivity can be essentially improved by coating its surface with a dense homogeneous ion-exchange film.

## Figures and Tables

**Figure 1 ijms-22-01415-f001:**
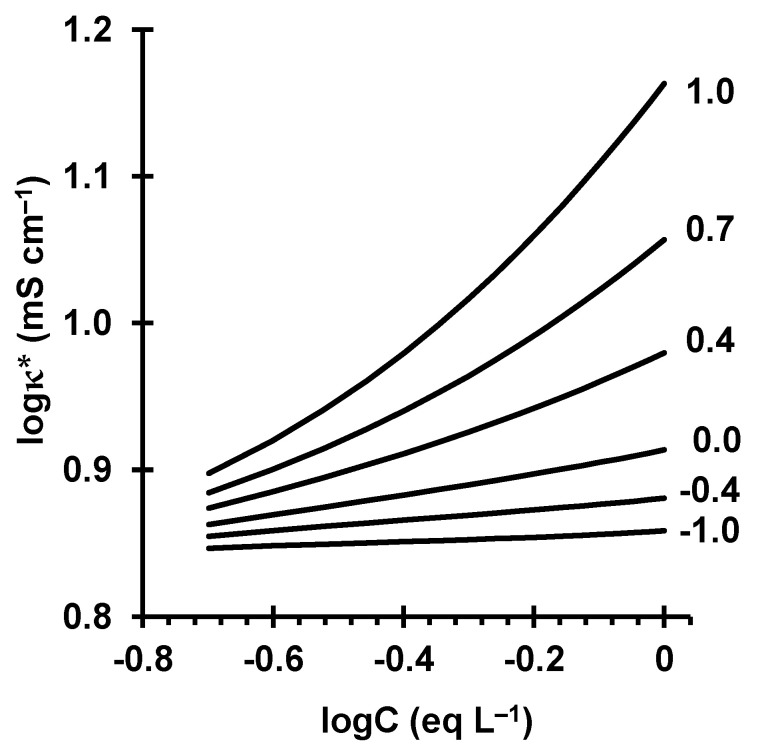
log$\kappa^{*}$—logC dependence at a fixed volume fraction of intergel solution, $f_{2}$, and different α (shown near the curves) calculated according to the microheterogeneous model [38] for the membrane parameters related to the AMH-PES membrane.

**Figure 2 ijms-22-01415-f002:**
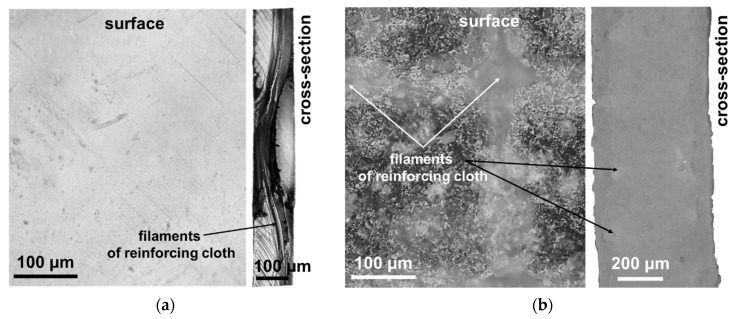
Optical images of surfaces and cross-sections of AMX (**a**), AMX-PES (**b**) membranes.

**Figure 3 ijms-22-01415-f003:**
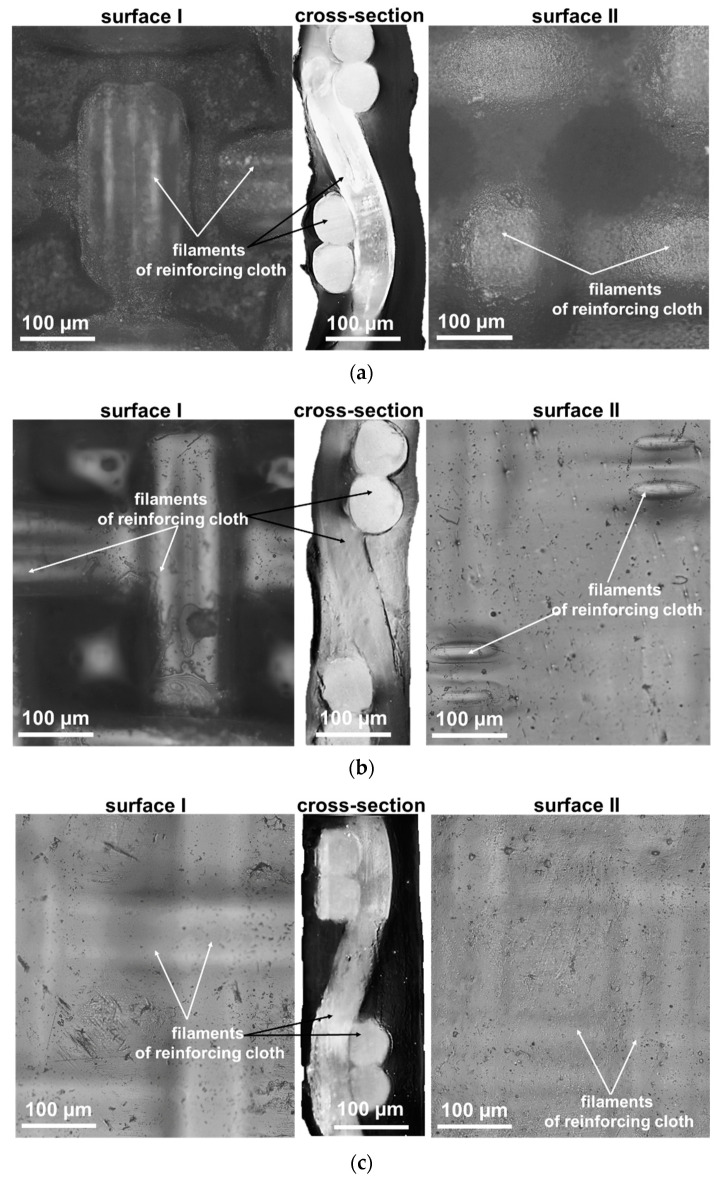
Optical images of surfaces and cross-sections of CJMA-3 (**a**), CJMA-6 (**b**) and CJMA-7 (**c**) membranes.

**Figure 4 ijms-22-01415-f004:**
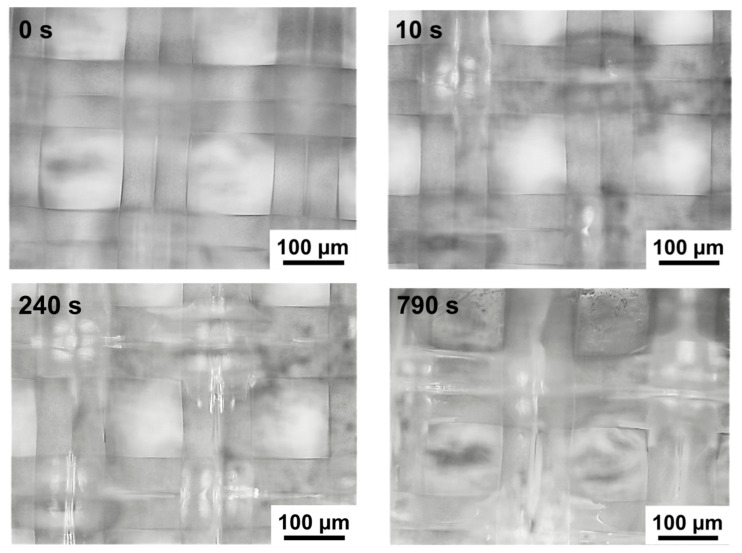
Optic visualization of the swollen CJMA-7 membrane drying process. The time of contact of the surface I with air in seconds is indicated on each video frame.

**Figure 5 ijms-22-01415-f005:**
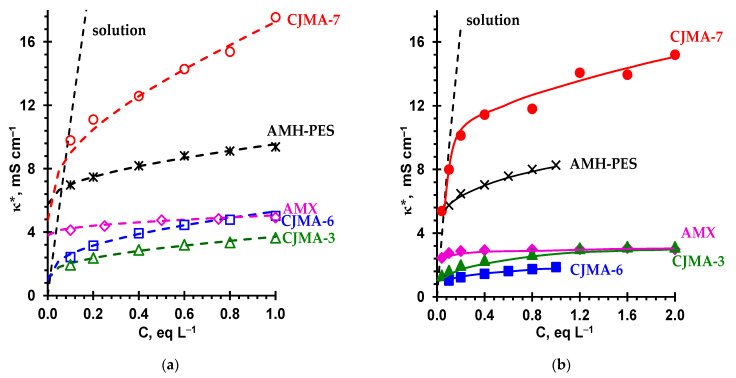
Concentration dependencies of the AEMs’ conductivity in NaCl (**a**) and Na_2_SO_4_ (**b**) solutions. Markers are experimental data; dashed lines are calculations using the microheterogeneous model [38]. The solid lines in (**b**) are drawn to guide the eye.

**Figure 6 ijms-22-01415-f006:**
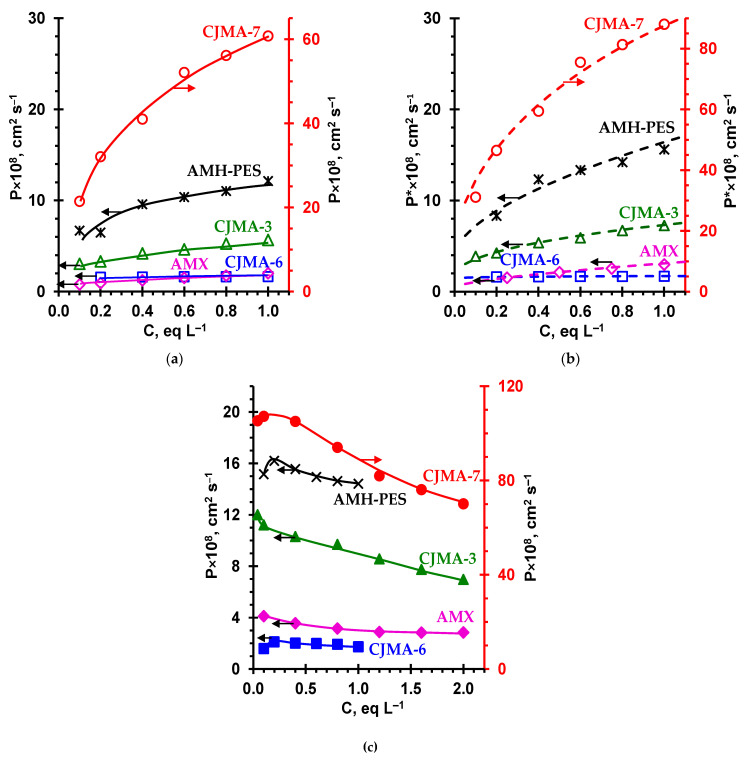
Concentration dependencies of the integral coefficient of diffusion permeability of AEMs in NaCl (**a**) and Na_2_SO_4_ (**c**) solutions as well as concentration dependency of the differential coefficient of diffusion permeability of AEMs in NaCl (**b**). Markers are experimental data; dashed lines are calculations using the microheterogeneous model [38]. The solid lines in (**a**) and (**c**) are drawn to guide the eye.

**Figure 7 ijms-22-01415-f007:**
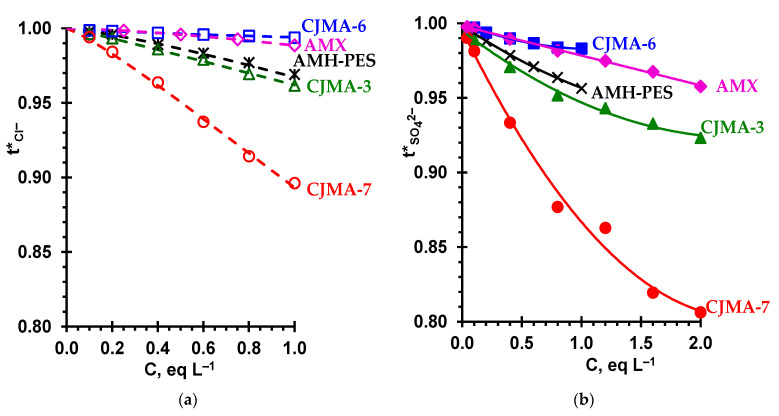
Concentration dependencies of counterion transport numbers in the AEM under study in NaCl (**a**) and Na_2_SO_4_ (**b**) solutions. The dashed lines in (**a**) are calculations using the microheterogeneous model [38]. The solid lines in (**b**) are drawn to guide the eye.

**Figure 8 ijms-22-01415-f008:**
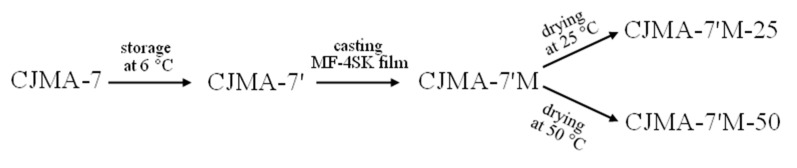
Scheme of the treatment applied to the CJMA-7 membrane.

**Figure 9 ijms-22-01415-f009:**
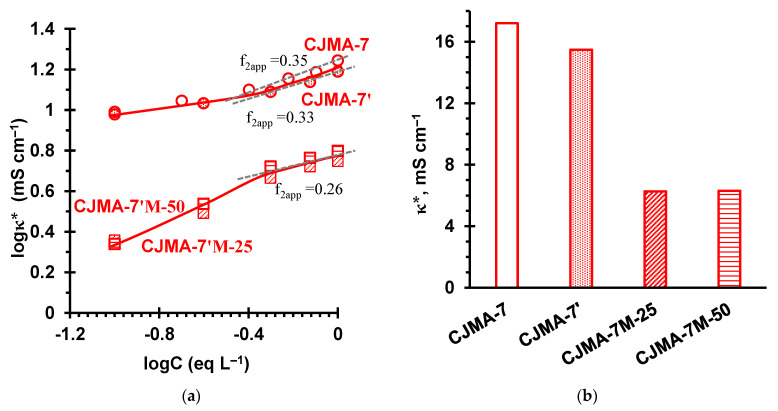
Concentration dependences of the electrical conductivity of the pristine membrane CJMA-7’ and modified samples CJMA-7’M in NaCl solutions (**a**) as well as comparison of the electrical conductivity of membranes under study in 1 eq L^−1^ NaCl (**b**). Indexes 25 and 50 denote the temperature (in °C) at which the modified sample was dried after casting a modifying film on its surface. The solid straight lines show the linear trend lines drawn to determine $f_{2app}$ in the range of elevated concentrations. The solid lines are drawn to guide the eye.

**Figure 10 ijms-22-01415-f010:**
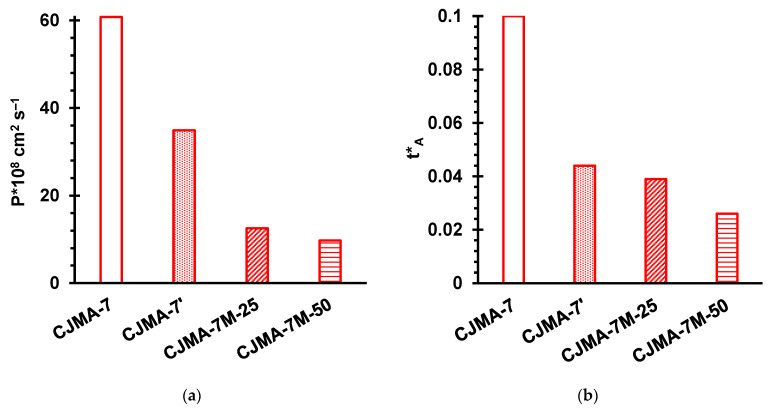
Diffusion permeability (**a**) and coion transport numbers (**b**) in the pristine membrane (CJMA-7) and modified samples. The data are obtained in 1.0 eq L^−1^ NaCl solution. Indexes 25 and 50 denote the temperature (in °C) at which the modified sample was kept after casting a modifying film on its surface.

**Table 1 ijms-22-01415-t001:** Materials and methods of manufacturing the membranes under study.

Membranes	Fixed Groups	Ion-Exchange Matrix	Inert Filler	Methods of Manufacturing	Reinforcing Cloth	Method of Embedding Reinforcing Cloth
AMX	Mainly –N^+^(CH_3_)_3_	DVB/PS	PVC	“Paste method”	PVC	Casting the paste on the cloth
AMH-PES			PE	Hot rolling	Ulester 32S	Hot rolling
CJMA-3		PVDF	Absent	Casting	PET	Casting the paste on the cloth
CJMA-6		Polyolefin				
CJMA-7						

**Table 2 ijms-22-01415-t002:** Some characteristics of the studied membranes. Our data are in bold without references.

Membranes	Thickness of AEM in 0.02 M NaCl, μm	Ion-Exchange Capacity of Swollen AEM, mmol/g_wet_	Water Content, g_H2O_/g_dry_, %	Concentration of Fixed Groups in Swollen AEM, mmol cm^−3^ H_2_O ^2^	Water Content, mol H_2_O/mol Fixed Groups	Density of Swollen AEM, g cm^−3^	Resistance of AEM in 0.5 M NaCl Solution, Ohm cm^2^
DVB/PS ion exchange matrix							
AMX	**135 ± 5** 141 ± 6 [58]	**1.25 ± 0.05** 1.25 [59]	**19 ± 1** 10−14 [60] 34 [61]	**7.7 ± 1.0**	**7.2 ± 1** 6.1 [58] 7.8 [62]	**1.22 ± 0.05** 1.10 [60]	**2.9 ± 0.5** 2.7 [63] 2.15 [61]
AMH-PES	**543 ± 10** 550 ± 3 [52]	**1.33 ± 0.01** 1.25 ± 0.08 [52] 0.86 [64]	**46 ± 5** 56 [65] 45.2 [64]	**4.2 ± 1.0**	**13.3 ± 0.7** 17.8 ± 1.0 [52]	**1.06 ± 0.05** 1.12 [66]	**6.4 ± 0.5** 6.1 [64] <7.5 [67] 7.66 [65]
PVDF or Polyolefin ion exchange matrix							
CJMA-3	**151 ± 5** 150 ± 20 ^1^ 130 [68] 250 [31]	**0.57 ± 0.05** 0.5−0.6 ^1^ 0.9 [68] 1.45 [31]	**17 ± 1** 15−20 ^1^ 26 [69] 35−45 [31]	**4.0 ± 1.0**	**14.0 ± 1.0**	**1.39 ± 0.05**	**5.1 ± 0.5** 4.0 ± 0.5 ^1^ 2 [68] 3−6 [31]
CJMA-6	**120 ± 3** 130 ± 10 [36]	**0.90 ± 0.05** 0.5−0.7 ^1^	**18 ± 1** 35−37 [36]	**6.0 ± 1.0**	**9.2 ± 1.0**	**1.32 ± 0.05**	**2.9 ± 0.5** 2.5−3.0 [53]
CJMA-7	**174 ± 10** 150 ± 20 ^1^ 200 [70]	**0.75 ± 0.05** 0.9−1.0 ^1^ 0.8 ± 1.0 [70]	**39 ± 2** 28−30 ^1^ 35−40 [70]	**2.6 ± 1.0**	**21.7 ± 1.0**	**1.13 ± 0.05**	**1.4 ± 0.5** 1.5 ± 1.8 ^1^ 1.5−2.5 [70]

^1^ The data were provided by the manufacturer [46]. ^2^ The number of millimoles of fixed groups per 1 cm^3^ of water in the membrane.

**Table 3 ijms-22-01415-t003:** Parameters of the microheterogeneous model for AEMs (found for the case of NaCl bathing solutions): the Cl^−^ counterion diffusion coefficient (${\bar{D}}_{Cl^{-}}$)) in the gel phase, ion-exchange capacity ($\bar{Q}$) of this phase, Gnusin’s parameter (G), structural parameter (α), apparent ($f_{2app}$) and “true” ($f_{2}$) volume fractions of the intergel solution of swollen membranes in NaCl and Na_2_SO_4_ solutions, as well as and the ratio of the SO_4_^2−^ to Cl^−^ diffusion coefficients in the gel phase. The value of $f_{2app}$ is found in the concentration range from 0.4 to 1.0 eq L^−1^.

Membranes	${\bar{D}}_{Cl^{-}}\;\times \;10^{6}$ cm^2^ s^−1^	$\frac{{\bar{D}}_{SO_{4}^{2-}}}{{\bar{D}}_{Cl^{-}}}$	$f_{2app}$		$f_{2}$	$\bar{Q}$, mmol cm^−3^	$G\;\times \;10^{3}$ mmol^−1^ cm^3^	α
			NaCl	Na_2_SO_4_	NaCl			
AMX	0.70	0.3 ± 0.2	0.10 ± 0.02 0.11 [40] 0.099 [60]	0.11 ± 0.02	0.03 ± 0.02	1.5 ± 0.1 1.39 [60]	0.32	0.39 ± 0.04
AMH-PES	1.20	0.4 ± 0.2	0.15 ± 0.02 0.14 [52]	0.17 ± 0.02	0.06 ± 0.02	1.5 ± 0.1 1.5 [52]	1.40	0.41 ± 0.04
CJMA-3	0.43	0.4 ± 0.2	0.27 ± 0.02	0.27 ± 0.02	0.12 ± 0.02	0.9 ± 0.1	0.17	0.28 ± 0.04
CJMA-6	0.32	0.7 ± 0.2	0.30 ± 0.02	0.29 ± 0.02	0.17 ± 0.02	1.4 ± 0.1		0.23 ± 0.04
CJMA-7	2.30	0.6 ± 0.2	0.35 ± 0.02	0.39 ± 0.02	0.18 ± 0.02	1.0 ± 0.1	8.4 ± 3	0.30 ± 0.04

## Data Availability

Not applicable.

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
