# Peer review of "Transport Characteristics of CJMAED™ Homogeneous Anion Exchange Membranes in Sodium Chloride and Sodium Sulfate Solutions"

_ijms, 2021, doi:10.3390/ijms22031415_

Round 1

Reviewer 1 Report

The reported study is performed under well-controlled conditions and the experimetal data obtained are reliable. The discussion is adequate and sound and the conclusions withdrawn relevant.  Although the study is not focused on techno-economical issues, comparing the costs (by giving at least rough estimates) of the three AEMs used would be of practical interest for the readers. Overall, I recommend acceptance of the manuscript after minor revision including its careful spell check and correcting the typing errors, which appear both in the text e.g.,  ow instead of low and in the units presentation mSm cm-1 instead of mS cm-1, etc. throughout the manuscript.  

Author Response

Dear Reviewer,

Many thanks to the Reviewer for the appreciation of our manuscript and for your efforts to improve it. We read the paper again and found some typing errors and insufficiently clear sentences. They have been corrected.

Authors’ response:

As for the cost of the membranes, it is confidential information of the manufacturer. However, we added the following sentence in line 233: “These membranes are less costly compared with other anion-exchange membranes on the world market.”

We can give you personally confidential information that the price of Chemjoy membranes is around $65/m2 in the China market, which is only one fifth of the price of Neosepta AMX/CMX membranes from Astom.

Corrections in the text of the manuscript are highlighted in yellow.

Reviewer 2 Report

Reviewer’s comment

This article is about modeling and analysis of CJMAED Homogeneous AEM according to electrolyte (species and concentration). Modeling results are well matched with experimental results. Also, it is quite interesting topic to researchers and engineers who want to understand the characteristics of AEMs. However, reviewer has some questions about modeling results as below: 

  • Please insert the units of all parameters in the text.
  • The CJMA membranes have reinforced substrate in the membrane matrix. However, author described CJMA membrane as homogeneous membrane. What is the definition of the meaning of homogeneous membrane? In my opinion, heterogeneous membrane does not contain a non-conductive substrate in the membrane matrix.
  • In line 80: What is the CJMAED? Is it a product name? Please describe it to help understanding.
  • Line 145: what is the ‘lg’? lg is not a common symbol. Please change it to ‘log’.
  • In a Figure 1 (and Eq. 1, 2), what is the unit of f1 and f2? According to the Eq. 1 and 2, dimension of the unit in Figure 1 should be reconsidered. If the unit is correct, please describe the meaning of the unit in detail.
  • Please describe Eq. 7 in detail. How Eq. 6 could be transformed to Eq. 7? What is the meaning of beta?
  • Modeling results are well matched with experimental results. How author improved accuracy (or precision) of the modeling results? What is the error of the modeling results? There is not a description about error.
  • In Figure 5 and 6, what is the meaning of the gradients change according to the C (eq. L-1)?
  • What is the meaning of “guide the eye” in Figure 5(b)? The results about modeling of Na2SO4 transport in membrane matrix by micro-heterogeneous model are not clear. To compare Figure 5a and 5b, micro-heterogeneous model should be adopted in Na2SO4 tests. Solid line in a Figure 5b leads misunderstanding of the result.
  • In Figure 5, What is the reason of missing points (of experiments and modeling) after 1.2 eq/L at AMH-PES?
  • What is the purpose of increasing permselectivity of CJMA-7?
  • What is the modeling result of modifying membrane by surface modification? Is there any difference between pristine membrane and modified membrane?

Author Response

Many thanks to the Reviewer for the interest and detailed analysis of our manuscript. All the comments are taken into account in the revised version. The answers to each comment are given below.

Corrections in the text of the manuscript are highlighted in yellow..

  1. The CJMA membranes have reinforced substrate in the membrane matrix. However, author described CJMA membrane as homogeneous membrane. What is the definition of the meaning of homogeneous membrane? In my opinion, heterogeneous membrane does not contain a non-conductive substrate in the membrane matrix.

Authors’ response:

The Reviewer is right. CJMA membranes really have reinforced substrate in the membrane matrix. However, the presence of a polymer-substrate does not prevent them from being classified as homogeneous membranes according to the generally accepted classification. The following fragment is added in lines 213-223:

“Conventionally, ion-exchange membranes are classified as homogeneous and heterogeneous. Heterogeneous membranes (which historically were produced earlier) are prepared by using powdered ion-exchange resin, which is mixed with a powder of a filler, such as polyethylene. The ion-exchange material (the particles of ion-exchange resin) is unevenly and sometimes discontinuously distributed in the membrane. Homogenous membranes are prepared by introducing an ion-exchange moiety directly into the structure of the constitutive polymer [45]. In this case, the ion-exchange material forms a continuous phase, which is relatively evenly distributed in the membrane. The reinforcing cloth (tissue, fabric) is added to both homogeneous and heterogeneous membranes. There are a few types of membranes (such as some marques of Nafion), which do not contain reinforcing cloth.”

  1. In line 80: What is the CJMAED? Is it a product name? Please describe it to help understanding.

Authors’ response:

The reviewer is absolutely right. CJMAED is the general product name for this type of membrane, emphasizing their possible application in electrodialysis. This clarification has been added to the text of the manuscript, line 226:

“The generic name of this type of membrane CJMAED on the manufacturer's website emphasizes their possible use in electrodialysis.”

  1. Line 145: what is the ‘lg’? lg is not a common symbol. Please change it to ‘log’.

Authors’ response:

Thanks to the Reviewer for the comment. We have replaced lg by log in the figures 1 and 9 as well as in the manuscript text.

  1. In a Figure 1 (and Eq. 1, 2), what is the unit of f1 and f2? According to the Eq. 1 and 2, dimension of the unit in Figure 1 should be reconsidered. If the unit is correct, please describe the meaning of the unit in detail.

Authors’ response:

f1 and f2 are the volume fractions of the gel phase (f1) and intergel spaces (f2). These parameters are dimensionless (since they are “fractions”), as it is indicated in line 129. To be clearer about the meaning of the parameters in Figure 1, we revised the legend:

« logk* - logC dependence at a fixed volume fraction of intergel solution, f2,…»

  1. Please describe Eq. 7 in detail. How Eq. 6 could be transformed to Eq. 7? What is the meaning of beta?.

Authors’ response:

Beta is defined in line 191: .

We have added the way to obtain Eq. (7) in lines 191 and 193:

« Eq. (7) is obtained from Eq. (6) as follows: .

The value of  can be found from the concentration dependence of P presented in bi-logarithmic plot (  vs. ). The slope of this dependence gives .»

  1. Modeling results are well matched with experimental results. How author improved accuracy (or precision) of the modeling results? What is the error of the modeling results? There is not a description about error.

Authors’ response:

Thank you for this question. The following fragment is added in lines 354-361:

«The values of these parameters were found by fitting the results of modelling to the experimental data. We have used the following features of the model behavior: when fitting the k* vs. С dependence, the f2 value controls the slope of the logk* vs. logС curve, while  shifts the curve up (when  increases) or down (when it decreases); when fitting the P* vs. С dependence, similarly, the slope and position of the logP* vs. logС may be changed by varying the values of parameters a and . Generally, there is a good agreement with the data known in the literature. As Figure 5 shows, the difference between the calculated and experimental results does not exceed the experimental error.»

  1. . In Figure 5 and 6, what is the meaning of the gradients change according to the C (eq. L-1)?

The meaning of the gradients change according to the C (eq. L-1) explanation is given in the text of the manuscript on the lines 333-338 and 444-447?

Authors’ response:

The causes of the concentration dependence of the conductivity and diffusion permeability are explained in lines 333-338 and in lines 444-447:

“With an increase in , the value of  changes insignificantly, mainly due to the sorption of coions. At the same time, the conductivity  of the intergel space (the central part of the pores) filled with a solution identical to the external one increases nearly proportional to . With increasing , the effect of the values of ,  and α on the membrane electrical conductivity rises. As mentioned above, the value of  increases with increasing concentration. It is due to increasing contribution of the intergel solution to the overall membrane conductivity; in addition, in the range of elevated concentrations, coions are more and more involved in the conductivity of the gel phase because of their increasing sorption by the gel.”

«Therefore, as predicted by the model, the diffusion permeability of these membranes increases with growth of external solution NaCl concentration in order AMX < AMH-PES according to the increase in the volume fraction  of the intergel solution.»

  1. What is the meaning of “guide the eye” in Figure 5(b)? The results about modeling of Na2SO4 transport in membrane matrix by micro-heterogeneous model are not clear. To compare Figure 5a and 5b, micro-heterogeneous model should be adopted in Na2SO4 Solid line in a Figure 5b leads misunderstanding of the result.

Authors’ response:

Thank you for your question. “…to guide the eye”  is a persistent expression (see e.g. J. Kamcev et al., Journal of Membrane Science 547 (2018) 123; http://dx.doi.org/10.1016/j.memsci.2017.10.024 ). “The line is drawn to guide the eye” means that a smooth line is drawn between the experimental points so that the trend can be easily traced by the reader.

Modeling of the characteristics of membranes in sodium sulfate solutions was not carried out, because the model does not take into account specific interactions of electrolytes in solution, while sulfates enter into such interactions. The presence of such interactions is evidenced by a sharp drop in the diffusion coefficient of Na2SO4 with an increase in the solution concentration (Figure S1, Supplementary materials).

  1. In Figure 5, What is the reason of missing points (of experiments and modeling) after 1.2 eq/L at AMH-PES.

Authors’ response:

As a rule, the concentration dependences of electrical conductivity are measured in the concentration range up to C≥10Сiso, which was done for membranes AMH-PES and CJMA-6. We decided to increase the range of Na2SO4 concentration in solutions for other membranes to make sure that in concentrated solutions specific interactions of sulfates with a solvent have a lesser effect on the electrical conductivity and to a greater extent affect the diffusion permeability of membranes.

  1. What is the purpose of increasing permselectivity of CJMA-7?

Authors’ response:

The higher the permselectivity of ion-exchange membrane, the higher the current efficiency and lower the energy consumption in an electrodialysis process. High coion transfer does not allow obtaining brines of high concentration. CJMA-7 has the lowest permselectivity among the studied membranes. We identified one of the reasons for the decrease in CJMA-7 permselectivity (the presence of extended macropores at the boundaries of the reinforcing cloth / ion-exchange matrix) and demonstrated one of the possible ways to eliminate this defect, which appears in the process of membrane reinforcement. In order to explain the aim of modification, we have added the following fragment in lines 108-112:

“In addition, we report the results of a surface modification of the CJMA-7 membrane, which significantly improved its counterion permselectivity, that is, decreased undesirable coion transfer. The latter reduces current efficiency of electrodialysis and does not allow obtaining brines of high concentration.”

  1. What is the modeling result of modifying membrane by surface modification? Is there any difference between pristine membrane and modified membrane?

Authors’ response:

Using a microheterogeneous model, it was shown that the volume fraction of the intergel space of the modified membranes decreased in comparison with the pristine membrane (see, please, figure 9a and lines 555-561). We revised this fragment of the text in order to link the changes in the membrane properties with the changes in its structure:

“As a result, the  value found from the concentration dependences of the electrical conductivity decreases from 0.33 (CJMA-7') to 0.28 (CJMA-7'M-25, CJMA-7'M-50). A decrease in the conductivity of the modified samples in the range of relatively high concentrations is explained by a decrease  in the volume fraction of a highly conductive electrically neutral solution in the membrane due to the structural changed caused by the modification of the pristine membrane. Note that the conductivity of the modified CJMA-7'M samples in 0.1 eq L-1 NaCl remains very high, about 6 mS cm-1, which is approximately equal to the conductivity of AMH-PES (Figure 4) and is noticeably higher than the conductivity of AMX (4 mS cm-1). The value of diffusion permeability decreased to a much greater extent (about 6 times, when comparing the CJMA-7 and CJMA-7M-50 membranes). The main cause is the decrease in the volume  fraction of the intergel electroneutral solution, which significantly reduces the possible routes of coion transport through the membrane. A strong decrease in the diffusion permeability leads to a three-fold decrease in the coion transport number (Figure 10b). ”

  1. Please insert the units of all parameters in the text?

Authors’ response:

It is done.

The dimensions for    f2, f2app, α,  , G, Q are given in the Table S1 (Supplimentary materials) and Tables 2, 3.

The dimensions for , P, P*, t* are given in the figures 5, 6, 7.

The dimensions for c and C are given in the lines 142-144:

“Here  (eq L-1) is the equivalent electrolyte concentration ( , where subscript 1 and A refer to the counterion and coion; ci (mol L−1) is the molar ion concentration)”.

Reviewer 2

Comments and Suggestions for Authors

This article is about modeling and analysis of CJMAED Homogeneous AEM according to electrolyte (species and concentration). Modeling results are well matched with experimental results. Also, it is quite interesting topic to researchers and engineers who want to understand the characteristics of AEMs.

Many thanks to the Reviewer for the interest and detailed analysis of our manuscript. All the comments are taken into account in the revised version. The answers to each comment are given below.

Corrections in the text of the manuscript are highlighted in yellow..

  1. The CJMA membranes have reinforced substrate in the membrane matrix. However, author described CJMA membrane as homogeneous membrane. What is the definition of the meaning of homogeneous membrane? In my opinion, heterogeneous membrane does not contain a non-conductive substrate in the membrane matrix.

Authors’ response:

The Reviewer is right. CJMA membranes really have reinforced substrate in the membrane matrix. However, the presence of a polymer-substrate does not prevent them from being classified as homogeneous membranes according to the generally accepted classification. The following fragment is added in lines 213-223:

“Conventionally, ion-exchange membranes are classified as homogeneous and heterogeneous. Heterogeneous membranes (which historically were produced earlier) are prepared by using powdered ion-exchange resin, which is mixed with a powder of a filler, such as polyethylene. The ion-exchange material (the particles of ion-exchange resin) is unevenly and sometimes discontinuously distributed in the membrane. Homogenous membranes are prepared by introducing an ion-exchange moiety directly into the structure of the constitutive polymer [45]. In this case, the ion-exchange material forms a continuous phase, which is relatively evenly distributed in the membrane. The reinforcing cloth (tissue, fabric) is added to both homogeneous and heterogeneous membranes. There are a few types of membranes (such as some marques of Nafion), which do not contain reinforcing cloth.”

  1. In line 80: What is the CJMAED? Is it a product name? Please describe it to help understanding.

Authors’ response:

The reviewer is absolutely right. CJMAED is the general product name for this type of membrane, emphasizing their possible application in electrodialysis. This clarification has been added to the text of the manuscript, line 226:

“The generic name of this type of membrane CJMAED on the manufacturer's website emphasizes their possible use in electrodialysis.”

  1. Line 145: what is the ‘lg’? lg is not a common symbol. Please change it to ‘log’.

Authors’ response:

Thanks to the Reviewer for the comment. We have replaced lg by log in the figures 1 and 9 as well as in the manuscript text.

  1. In a Figure 1 (and Eq. 1, 2), what is the unit of f1 and f2? According to the Eq. 1 and 2, dimension of the unit in Figure 1 should be reconsidered. If the unit is correct, please describe the meaning of the unit in detail.

Authors’ response:

f1 and f2 are the volume fractions of the gel phase (f1) and intergel spaces (f2). These parameters are dimensionless (since they are “fractions”), as it is indicated in line 129. To be clearer about the meaning of the parameters in Figure 1, we revised the legend:

« logk* - logC dependence at a fixed volume fraction of intergel solution, f2,…»

  1. Please describe Eq. 7 in detail. How Eq. 6 could be transformed to Eq. 7? What is the meaning of beta?.

Authors’ response:

Beta is defined in line 191: .

We have added the way to obtain Eq. (7) in lines 191 and 193:

« Eq. (7) is obtained from Eq. (6) as follows: .

The value of  can be found from the concentration dependence of P presented in bi-logarithmic plot (  vs. ). The slope of this dependence gives .»

  1. Modeling results are well matched with experimental results. How author improved accuracy (or precision) of the modeling results? What is the error of the modeling results? There is not a description about error.

Authors’ response:

Thank you for this question. The following fragment is added in lines 354-361:

«The values of these parameters were found by fitting the results of modelling to the experimental data. We have used the following features of the model behavior: when fitting the k* vs. С dependence, the f2 value controls the slope of the logk* vs. logС curve, while  shifts the curve up (when  increases) or down (when it decreases); when fitting the P* vs. С dependence, similarly, the slope and position of the logP* vs. logС may be changed by varying the values of parameters a and . Generally, there is a good agreement with the data known in the literature. As Figure 5 shows, the difference between the calculated and experimental results does not exceed the experimental error.»

  1. . In Figure 5 and 6, what is the meaning of the gradients change according to the C (eq. L-1)?

The meaning of the gradients change according to the C (eq. L-1) explanation is given in the text of the manuscript on the lines 333-338 and 444-447?

Authors’ response:

The causes of the concentration dependence of the conductivity and diffusion permeability are explained in lines 333-338 and in lines 444-447:

“With an increase in , the value of  changes insignificantly, mainly due to the sorption of coions. At the same time, the conductivity  of the intergel space (the central part of the pores) filled with a solution identical to the external one increases nearly proportional to . With increasing , the effect of the values of ,  and α on the membrane electrical conductivity rises. As mentioned above, the value of  increases with increasing concentration. It is due to increasing contribution of the intergel solution to the overall membrane conductivity; in addition, in the range of elevated concentrations, coions are more and more involved in the conductivity of the gel phase because of their increasing sorption by the gel.”

«Therefore, as predicted by the model, the diffusion permeability of these membranes increases with growth of external solution NaCl concentration in order AMX < AMH-PES according to the increase in the volume fraction  of the intergel solution.»

  1. What is the meaning of “guide the eye” in Figure 5(b)? The results about modeling of Na2SO4 transport in membrane matrix by micro-heterogeneous model are not clear. To compare Figure 5a and 5b, micro-heterogeneous model should be adopted in Na2SO4 Solid line in a Figure 5b leads misunderstanding of the result.

Authors’ response:

Thank you for your question. “…to guide the eye”  is a persistent expression (see e.g. J. Kamcev et al., Journal of Membrane Science 547 (2018) 123; http://dx.doi.org/10.1016/j.memsci.2017.10.024 ). “The line is drawn to guide the eye” means that a smooth line is drawn between the experimental points so that the trend can be easily traced by the reader.

Modeling of the characteristics of membranes in sodium sulfate solutions was not carried out, because the model does not take into account specific interactions of electrolytes in solution, while sulfates enter into such interactions. The presence of such interactions is evidenced by a sharp drop in the diffusion coefficient of Na2SO4 with an increase in the solution concentration (Figure S1, Supplementary materials).

  1. In Figure 5, What is the reason of missing points (of experiments and modeling) after 1.2 eq/L at AMH-PES.

Authors’ response:

As a rule, the concentration dependences of electrical conductivity are measured in the concentration range up to C≥10Сiso, which was done for membranes AMH-PES and CJMA-6. We decided to increase the range of Na2SO4 concentration in solutions for other membranes to make sure that in concentrated solutions specific interactions of sulfates with a solvent have a lesser effect on the electrical conductivity and to a greater extent affect the diffusion permeability of membranes.

  1. What is the purpose of increasing permselectivity of CJMA-7?

Authors’ response:

The higher the permselectivity of ion-exchange membrane, the higher the current efficiency and lower the energy consumption in an electrodialysis process. High coion transfer does not allow obtaining brines of high concentration. CJMA-7 has the lowest permselectivity among the studied membranes. We identified one of the reasons for the decrease in CJMA-7 permselectivity (the presence of extended macropores at the boundaries of the reinforcing cloth / ion-exchange matrix) and demonstrated one of the possible ways to eliminate this defect, which appears in the process of membrane reinforcement. In order to explain the aim of modification, we have added the following fragment in lines 108-112:

“In addition, we report the results of a surface modification of the CJMA-7 membrane, which significantly improved its counterion permselectivity, that is, decreased undesirable coion transfer. The latter reduces current efficiency of electrodialysis and does not allow obtaining brines of high concentration.”

  1. What is the modeling result of modifying membrane by surface modification? Is there any difference between pristine membrane and modified membrane?

Authors’ response:

Using a microheterogeneous model, it was shown that the volume fraction of the intergel space of the modified membranes decreased in comparison with the pristine membrane (see, please, figure 9a and lines 555-561). We revised this fragment of the text in order to link the changes in the membrane properties with the changes in its structure:

“As a result, the  value found from the concentration dependences of the electrical conductivity decreases from 0.33 (CJMA-7') to 0.28 (CJMA-7'M-25, CJMA-7'M-50). A decrease in the conductivity of the modified samples in the range of relatively high concentrations is explained by a decrease  in the volume fraction of a highly conductive electrically neutral solution in the membrane due to the structural changed caused by the modification of the pristine membrane. Note that the conductivity of the modified CJMA-7'M samples in 0.1 eq L-1 NaCl remains very high, about 6 mS cm-1, which is approximately equal to the conductivity of AMH-PES (Figure 4) and is noticeably higher than the conductivity of AMX (4 mS cm-1). The value of diffusion permeability decreased to a much greater extent (about 6 times, when comparing the CJMA-7 and CJMA-7M-50 membranes). The main cause is the decrease in the volume  fraction of the intergel electroneutral solution, which significantly reduces the possible routes of coion transport through the membrane. A strong decrease in the diffusion permeability leads to a three-fold decrease in the coion transport number (Figure 10b). ”

  1. Please insert the units of all parameters in the text?

Authors’ response:

It is done.

The dimensions for    f2, f2app, α,  , G, Q are given in the Table S1 (Supplimentary materials) and Tables 2, 3.

The dimensions for , P, P*, t* are given in the figures 5, 6, 7.

The dimensions for c and C are given in the lines 142-144:

“Here  (eq L-1) is the equivalent electrolyte concentration ( , where subscript 1 and A refer to the counterion and coion; ci (mol L−1) is the molar ion concentration)”.
